# Dynamic Changes in the Bacterial Community During the Fermentation of Traditional Chinese Fish Sauce (TCFS) and Their Correlation with TCFS Quality

**DOI:** 10.3390/microorganisms7090371

**Published:** 2019-09-19

**Authors:** Fangmin Du, Xiaoyong Zhang, Huarong Gu, Jiajia Song, Xiangyang Gao

**Affiliations:** 1Guangdong Provincial Key Laboratory of Nutraceuticals and Functional Foods, College of Food Science, South China Agricultural University, Guangzhou 510642, China; dufangmin1995@163.com (F.D.); guhuarong1996@163.com (H.G.); geffsong@163.com (J.S.); 2Joint Laboratory of Guangdong Province and Hong Kong Region on Marine Bioresource Conservation and Exploitation, College of Marine Sciences, South China Agricultural University, Guangzhou 510642, China; zhangxiaoyong@scau.edu.cn

**Keywords:** fish sauce, bacterial community, *Halanaerobium*, *Tetragenococcus*

## Abstract

This study revealed for the first time the dynamic changes of the bacterial community during the fermentation of traditional Chinese fish sauce (TCFS) using high-throughput sequencing. In the early phase of TCFS fermentation, *Shewanella* (approximately 90%) within *Proteobacteria* was the dominant bacteria. Then, *Halanaerobium* (3%–86%) within *Firmicutes* rapidly replaced *Shewanella* as the dominant genus until the 12th month. *Lactococcus* (3.31%) and *Bacillus* (45.56%) belonging to *Firmicutes* were detected abundantly in the 3rd and 9th months after fermentation, respectively. In the late phase (12–15 months), *Tetragenococcus* within *Firmicutes* replaced *Halanaerobium* as the most dominant bacteria (29.54%). Many other genera including *Pseudomonas*, *Psychrobacter*, *Tissierella, Carnobacterium* and *Gallicola* were abundantly present in the 15th month after fermentation. Furthermore, the relationships between the bacterial community and major functional substances of TCFS, including amino nitrogen (AAN), free amino acids (FAAs), total soluble nitrogen (TSN), and trimethylamine (TMA), were investigated by partial least squares regression (PLSR). *Tetragenococcus* was positively correlated with the formation of TMA, while *Halanaerobium* showed the opposite result, suggesting that *Tetragenococcus* might be a good starter for TCFS fermentation. These results contribute to our knowledge about bacterial participation in the process of TCFS fermentation and will help improve the quality of fermented seafood.

## 1. Introduction

Fermented food plays an important role in human diet and is widely consumed all over the world. In recent years, it has become increasingly popular because of its benefits to human health [1,2]. Fish sauce, also known as fish soy sauce, is mainly made by mixing low-value fish with a certain proportion of sea salt (fish:salt = 3:1 or 2:1) as raw material [3] and allowing it naturally ferment over months or years. Fish sauce has not only unique flavors but also rich nutrients, including all essential amino acids, taurine vitamins, trace elements, and a large number of bioactive peptides [4], which are formed by decomposing the protein and fat of raw fish in the fermentation process with the combined action of proteases contained in the fish body and various microorganisms. Fish sauce is a traditional characteristic condiment in Southeast Asia [5,6], being a main source of dietary protein for residents of various countries and regions. In the past few decades, the fish sauce industry in China has developed rapidly, with a domestic output of more than 100,000 tons per year. Guangdong Chaoshan fish sauce is one of the typical representatives of traditional Chinese fish sauce (TCFS).

The quality and flavor of TCFS are closely related to the microbial community. The core strains of fish sauce fermentation could be selected by defining the rule of bacterial community succession in fish sauce fermentation and analyzing the correlation between bacterial community and fish sauce quality. In particular fermentation stages, beneficial bacteria can be added to accelerate the fermentation process, shorten the fermentation period, and improve product quality. It is of great significance to monitor and standardize the traditional technology as well as the quality and safety of TCFS to clarify the microbial community structure and its changes and functions in the process of fish sauce fermentation. It has been reported that many halophilic or salt tolerant bacteria, including *Filobacillus*, *Bacillus*, *Micrococcus*, *Staphylococcus*, *Virgibacillus*, *Pseudomonas*, *Halobacillus,* and *Halococcus* [7,8,9,10,11], have been isolated from fish sauce using culture-dependent approaches. In addition, *Virgibacillus* sp. SK33 [12], *Bacillus* subtilis CN2 [13], and V. *halodenitrificans* SK1-3-7 [14] have been isolated from fish sauce and have high protease activity.

Recently, an increasing number of studies have revealed that approximately 99% of the microbial community in the environment is uncultured, which limits the applications of traditional culture methods. With the development of molecular biology technology, the non-culture method based on high-throughput sequencing technology allows a comprehensive and accurate understanding of microbial community structures in the environment and provides a new way to study their diversity and function [15,16,17]. High-throughput sequencing is mainly performed through sequencing by synthesis (SBS) and the exclusive reversible termination of chemical reactions; therefore, large amounts of data can be obtained in a short time, with high throughput, high accuracy, and low cost. High-throughput sequencing technology has been widely used in fermented foods, such as cheese [18,19], beer [20], wine [21,22], vinegar [23], fermented soybean products [24,25,26], sausage [27], fermented vegetables [28,29], fermented beverages [30], and fermented fish [31]. Recently, high-throughput sequencing technology was used to analyze the bacteria in South Korean fish sauce, where *Halanaerobium* and *Tetragenococcus* were revealed as the dominant microbes. However, the microbial structure of fermented food tends to be affected by the region [25], and the structure of the bacterial community during the fermentation process of TCFS has not been reported. Therefore, research on the influence of the microflora on the quality of fish sauce needs to be further studied.

In this study, a high-throughput sequencing technology was used to reveal the dynamic changes in the bacterial community during the natural fermentation of TCFS. The correlation between the bacterial community and the major functional substrates of TCFS were investigated using the SIMCA-P package. The results will broaden our knowledge of microorganisms participating in the fermentation process and guide the quality control of fermented fish foods.

## 2. Materials and Methods

### 2.1. Sample Collection and Analysis

TCFS samples were provided by Shantou Fish Sauce Factory Co. LTD (shantou, China) and made via the traditional fish sauce process. TCFS with a salt concentration of approximately 30% (*w*/*v*) was prepared in triplicate using anchovies caught from Jia Zi and Lu Feng according to a traditional manufacturing method. The fresh anchovies and solar salts (Shantou, China) were dispensed into fermentation pools in portions of 21 t and 8 t, respectively. Then, 30 t of a 30% (*w*/*v*) solar salt solution was added to each pool to completely cover the anchovies and was stirred twice a month. Samples were taken every three months until fermentation reached the 15th month.

Considering the homogeneity of the samples, a 5-point sampling method was adopted for sampling. That is, samples were prepared by pooling equal amounts of mixture from each of the four corners and middle point of the fermentation pool. An equal amount of sample was taken from the surface, middle, and bottom of each point (Figure 1). All the fish sauce fermentation samples at the 5 points were mixed, and three parallel operations were conducted to obtain a total of 18 samples.

The fish bodies and precipitates in the fermentation broth were homogenously mixed and centrifuged for precipitation (5000 r/min for 10 min at 20 °C) and stored at −80 °C prior to microbial community analyses. All samples were filtered to clear any brown liquid and stored at −20 °C before physical and chemical indices were measured.

The concentrations of amino nitrogen (AAN) and trimethylamine (TMA) of the fish sauce were determined according to the methods described by Beddows et al. [32] and Dissaraphong et al. [33], respectively. The concentration of total soluble nitrogen (TSN) of each sample was determined using the Kjeldahl method according to official AOAC standards [5]. The pH value was measured directly with a digital pH meter (PB-10, Sartorius Group, Gottingen, Germany).

The concentration of free amino acids (FAAs) was measured according to the method described by Zeng et al. [34], with slight modifications. Briefly, according to the concentration of amino acids, deionized water was used to dilute the fish sauce sample by an appropriate multiple, which was kept for 30 min at 4 °C. Then, 1 mL was removed, 1 mL sulfonate-salicylic acid (50 g/L) was added, and the mixture was kept for 1 h at 4 °C. Subsequently, 0.5 mL EDTA (50 mg/mL) was added, and the sample was kept overnight at 4 °C. Then, the sample was centrifuged (12000 r/min, 4 °C, 15 min), the supernatant was taken, and the filtrate was obtained using a water membrane (0.22 μm).

Subsequently, the filtrate was used for analysis of the amino acid composition using an automatic amino acid analyzer (S-433D SYKAM, Munich, Germany) with a column (LCA K07/Li). The main working parameters of the instrument were as follows: mobile phase, lithium citrate A = pH 2.90, B = pH 4.20, C = pH 8.00; detection wavelength, 570 nm + 440 nm; flow rate, elution pump 0.45 mL/min + derivative pump 0.25 mL/min; temperature, 38−74 °C gradient temperature rise. The fish sauce samples were identified as xm, where “x” designated the sampling month during the fish sauce fermentation period.

### 2.2. DNA Extraction and Illumina MiSeq Sequencing

Total DNA of the TCFS samples was extracted using the E.Z.N.A.^®^ Soil DNA Kit (Omega Bio-tek, Norcross, GA, US) according to the manufacturer’s protocols. The final DNA concentration and purification were determined with a Nano Drop 2000 UV-vis spectrophotometer (Thermo Scientific, Wilmington, DE, USA), and DNA quality was checked by 1% agarose gel electrophoresis.

The forward primer 338F (5′-ACTCCTACGGGAGGCAGCAG-3′) and the reverse primer 806R (5′-GGACTACHVGGGTWTCTAAT-3′) were used to amplify the hypervariable regions of the bacterial 16S rRNA gene (V3-V4), which were extracted from fish sauce samples. PCR amplification was performed in triplicate in 20 μL reactions with 4 μL of 5 × Fast Pfu Buffer, 0.8 μL of each primer (5 μM), 2 μL of 2.5 mM dNTPs, 0.4 μL of Fast Pfu Polymerase, and 10 ng of template DNA, conducted using the following conditions: 3 min of denaturation at 95 °C; 27 cycles of 30 s at 95 °C, 30 s for annealing at 55 °C, and 45 s for elongation at 72 °C; and a final extension at 72 °C for 10 min, with a thermocycler PCR system (Gene Amp 9700, ABI, Chino, CA, USA). PCR products were purified using the AxyPrep DNA Gel Extraction Kit (Axygen Biosciences, Union City, CA, USA) and QuantiFluor™-ST (Promega, Madison, WI, USA) according to the manufacturer’s protocol. Purified amplicons were pooled in equimolar amounts to construct the 16S rRNA sequencing libraries and then paired-end sequenced (2 × 300) on an Illumina MiSeq platform (Illumina, San Diego, CA, USA) according to the standard protocols by Majorbio Bio-Pharm Technology Co. Ltd. (Shanghai, China). 

### 2.3. Sequencing Processing and Data Analysis

To minimize the influence of poor sequence quality and sequencing errors, the raw sequences were filtered with the Trimmomatic software (Version 0.36). Sample-specific sequences were assigned using their barcodes allowing 2 nucleotide mismatching; reads containing ambiguous bases and average quality scores below 20 were removed. Sequences that overlapped more than 10 bp were merged according to the overlap sequence with the FLASH software.

Operational taxonomic units (OTUs) were clustered with a 97% similarity cut-off using UPARSE (version 7.1 http://drive5.com/uparse/), and chimeric sequences were identified and removed using UCHIME. The taxonomy of each 16S rRNA high-quality sequence was processed with the RDP Classifier algorithm (http://rdp.cme.msu.edu/) against the Silva (SSU123) 16S rRNA database using a confidence threshold of 70%. Chao1 [35] and Shannon [36] indices and coverage values were calculated with the Mothur software (version 1.30.1). A rarefaction curve [37] was also generated within the Mothur program based on alpha diversity indices. Hierarchical clustering was carried out according to the beta diversity distance matrix, which was calculated with the QIIME software (version 1.9.0), and UPGMA was used to construct the tree structure.

### 2.4. Statistical Analysis

All determinations were carried out in triplicate, and the results are reported as means ± SD. Regression coefficients were calculated using the SIMCA-P package (SIMCA-P 11.5, Umea, Sweden) on the physical and chemical indices (PACI) and the bacterial community abundance to investigate the correlation between PACI and the relative abundance of the bacterial community during fish sauce fermentation.

### 2.5. Nucleotide Sequence Accession Numbers

The raw sequencing data associated with this study were uploaded to the NCBI Sequence Read Archive (SRA) database (Accession Number: SRR9618343-9618360).

## 3. Results

### 3.1. Physical and Chemical Indicator Changes in Fish Sauce Samples During Fermentation

In this study, various physicochemical indexes were monitored (Table 1), and PLSR was used to analyze the change trend of physicochemical indexes in different fermentation periods of fish sauce (Figure 2). The results showed that with the extension of fermentation time, the concentration of TSN, AAN, and FAAs increased, which are indicators reflecting the decomposition of fish protein and the main indicators of fish sauce product classification. In the 15th month, TSN, AAN, and FAA contents reached 12 mg/mL, 7.7 mg/mL, and 30.6 mg/mL, respectively. This indicated that fish protein is increasingly degraded by microorganisms. Meanwhile, the pH is an important indicator reflecting the process of lactic acid fermentation, and it can be seen in Figure 2 that pH presents a declining trend, and reaches 5.55 at 15^th^ month. As a whole, the concentration of TMA presented a downward trend. TMA is formed by reducing TMAO in the decomposition process of fish and meat due to bacterial action. The TMA content first increases, then decreases, reaching 140 μg/mL in the 6th month, and then decreases further, reaching 94 μg /mL in the 15th month.

### 3.2. Illumina MiSeq Sequencing Data Statistics

After quality control processes filtered out reads containing incorrect primer or barcode sequences and sequences that were shorter than 300 nucleotides or with more than one ambiguous base, a total of 1032674 effective sequences with average sequences of 448 ± 3 bp and 57371 ± 4401 sequences per sample were obtained. In addition, a total of 6731 OTUs were obtained, 374 ± 13 per sample, which were clustered by 97% sequence similarity (Table 1). When a rarefaction analysis was carried out to determine whether all the OTUs present in the datasets had been sufficiently recovered in the pyrosequencing study, all rarefaction curves showed a similar pattern of becoming gradually stable and reached a saturation phase rapidly, suggesting that the majority of the bacterial communities present in the fish sauce samples were identified (Figure 3).

Alpha diversity is the result of an analysis of species diversity in an environment, the Chao index reflects community richness, and Shannon and Simpson indexes reflect community diversity. The alpha-diversity data showed that the Chao diversity index and the Shannon diversity index varied from 254 ± 70 to 763 ± 36, and from 0.57 ± 0.27 to 3.45 ± 0.28, respectively, during fermentation (Table 2). The richness and diversity of the bacterial community in fish sauce increased gradually with the extension of fermentation time. In addition, Good’s coverage for the samples with the coverage index was above 0.99 for all samples.

### 3.3. Analysis of Bacterial Community Composition in the Fish Sauce Fermentation Process

A total of 6731 OTUs obtained from our fish sauce samples contained a rich bacterial community from 24 phyla and 558 genera. There were 2 major bacterial phyla in the fish sauce, including *Firmicutes* (75.22 ± 1.16%) and *Proteobacteria* (22.62 ± 1.50%). A small proportion of bacterial phyla, including *Bacteroidetes*, *Actinobacteria*, *Fusobacteria*, and *Cyanobacteria* (Figure 4a), was detected in the fish sauce. At the genus level, nine major genera including *Halanaerobium* (52.17 ± 2.93%), *Shewanella* (14.86 ± 0.43%), *Bacillus* (9.68 ± 1.09%), *Tetragenococcus* (5.96 ± 1.57%), *Vibrio* (2.04 ± 0.07%), *Gallicola* (1.47 ± 0.31%), *Photobacterium* (1.19 ± 0.62%), *Pseudomonas* (1.14 ± 0.24%), and *Psychrobacter* (1.01 ± 0.10%) were abundantly found in the fish sauce. In addition, most genera were less than 1% abundant and were classified as “others” (Figure 4b).

### 3.4. Beta Diversity Analysis

To study the similarity and difference in the species composition structure of the different samples, a hierarchical clustering tree was constructed with the UPGMA algorithm, and Bray-Curtis algorithm was used to calculate the sample distance, which revealed three different clusters (Figure 5): group 1–0 months; group 2–15 months; group 3–3 months, 6 months, 9 months, and 12 months. The results showed that the bacterial community structures of the 3rd, 6th, 9th, and 12th months were slightly different from those of the early and late (15th month) fermentation stages.

At the phylum level (Figure 6a), *Proteobacteria* decreased rapidly from fermentation until the 9th month; the species abundance decreased from 93.65% to 5.71% and recovered slightly, reaching 21.58% in the 15th month of fermentation. In contrast, *Firmicutes* rose rapidly from fermentation, and the species abundance increased until the 9th month of fermentation, rising from 6.05% to 93.11%, and then decreased slightly to 72.78% in the 15th month of fermentation, replacing *Proteobacteria* as the dominant bacteria in the fermentation process.

At the genus level (Figure 6b), *Shewanella* (approximately 90%) was the most dominant bacteria in the early phase of TCFS fermentation (0–3 months). Then, *Halanaerobium* (3%–86%) rapidly replaced *Shewanella* as the dominant genus until the 12th month of fermentation. In the late phase of TCFS fermentation (12–15 month), *Tetragenococcus*, within Firmicutes, replaced *Halanaerobium* as the most dominant bacteria (approximately 29.54%). Other genera within *Proteobacteria* and *Firmicutes*, including *Pseudomonas*, *Psychrobacter*, *Tissierella*, *Carnobacterium,* and *Gallicola*, were abundantly present in the 15th month of fermentation.

### 3.5. Correlation Analysis of Environmental Factors

The correlation between bacterial community and metabolites was studied using the PLSR method based on different fermentation periods of fish sauce samples. *Halanaerobium*, *Tetragenococcus*, *Vibrio*, *Pseudomonas*, *Pseudoalteromonas*, *Psychrobacter,* and *Gallicola* were selected for correlation analysis with the above physical and chemical indicators (Figure 7). The PLSR model extracted three principal components, the crossover validity was 0.90, and the amount of interpretation reached 0.66. This model explained our data well, and the results showed that all genera showed positive correlations with FAAs, TSN, and AAN (Figure 7: FAA, TSN, AAN), indicating that each genus had positive effects on the accumulation of amino acids, among which *Halanaerobium, Tetragenococcus,* and *Pseudoalteromonas* had high correlations. *Tetralococcus* and *vibrio* were negatively correlated with pH, while *Halanaerobium* and *Pseudoalteromonas* were positively correlated with pH (Figure 7: pH). *Halanaerobium* and *Pseudoalteromonas* play a promoting role in the formation of trimethylamine, while *Tetragenococcus* showed the opposite. There was no significant correlation between *Vibrio* and TMA (Figure 7: TMA).

## 4. Discussion

The formation of unique flavors and nutrients in TCFS is closely related to the activity of microorganisms. Few studies have been carried out on the microbial community in the fermentation process of TCFS. Meanwhile, clarifying the structure of the bacterial community in TCFS fermentation is helpful to reduce the potential risk of natural fermentation to human health [38,39]. This study revealed for the first time the dynamic succession of the bacterial community in the process of natural fermentation of TCFS using high-throughput sequencing and examined the influence of the flora on the fish sauce quality. The dominant bacteria in fish sauce fermentation were *Firmicutes* (75.22%) (mainly including *Halanaerobium*, *Bacillus*, and *Tetragenococcus*) and *Proteobacteria* (22.62%) (mainly including *Shewanella*) (Figure 4), which was consistent with previous results stating that most of the bacteria in fermented food are *Firmicutes* and *Proteobacteria.* However, due to the differences in regional characteristics and raw materials, the abundance ratio was different, which also reflected the restriction effect of the salt environment on the microbial community composition. In addition, we also found that some bacteria were not consistent with the NCBI database (mainly including unclassified *Clostridiales*) (Figure 6), which suggests that many new bacterial species may be involved in the fermentation of TCFS.

During the fermentation of fish sauce, the structure of the microbial community changed constantly. Our data showed that the diversity and richness of the bacterial community increased gradually with the extension of fermentation time, where the Chao and Shannon diversity indexes varied from 254 ± 70 to 763 ± 36, and from 0.57 ± 0.27 to 3.45 ± 0.28, respectively (Table 1). At the phylum level, the abundance of *Proteobacteria* in the process of TCFS fermentation displayed a trend of early decrease and later increase, while *Firmicutes* displayed the opposite trend. The abundance of *Proteobacteria* and *Firmicutes* reached 21.58% and 72.78% after 15 months of fermentation, respectively (Figure 6). *Firmicutes* and *Proteobacteria* were the most dominant phyla during the entire fish sauce fermentation period (Figure 6), which was consistent with previously reported results for other fermented salted food [40,41,42], shrimp paste [43], and soybean paste [44].

Known as a major putrefying bacterium in fish and reductase-producing bacterium [45,46,47], *Shewanella* was initially predominant (90%, Figure 6). However, it declined rapidly after the 3rd month of fermentation, suggesting that marination may contribute to the production of safe seafood [40]. *Halanaerobium*, *Lactococcus,* and *Bacillus* appeared successively in the fermentation process and became the dominant bacteria (Figure 6). In 1985, Schleifer et al. proposed *Lactococcus* to establish a new genus of bacteria, facultative anaerobic bacteria, which could ferment carbohydrates and produce acids. In addition, it was reported that *Lactococcus* could cause various human infections [48,49], but it only appeared in the 3rd month of fermentation and then disappeared rapidly. *Bacillus* is a gram-negative bacterium that is strictly aerobic or facultative anaerobic and can ferment glucose and amino acids to produce lactic acid. However, it disappeared rapidly after its appearance in the 9th month of fermentation.

In the process of fermentation, *Halanaerobium* was still the dominant bacterium until the 12th month of fermentation, and then the abundance of *Tetragenococcus* rapidly increased to 29.54% (Figure 6), replacing *Halanaerobium* as the most predominant genus. In contrast to other fermented seafood, *Staphylococcus* did not appear [37]. This result was consistent with the results in Figure 7, which showed a positive correlation between *Halanaerobium* and TMA (Figure 7: TMA). It was reported that members of *Halanaerobium* are responsible for the production of acetate, butyrate, and biogenic amines through the fermentation of monosaccharides, amino acids, and glycerol [40,41,42].

*Tetragenococcus* can tolerate high salt amounts, has a diverse protease activity, and is often used to improve the total amino acid and tasty amino acid content in fermented foods and condiments [50,51]. Our data showed that *Tetragenococcus* was positively correlated with FAAs and negatively correlated with pH and TMA (Figure 7: FAAs, TMA), indicating that this genus had the potential not only to produce acid but also to reduce harmful substances (ammonia and amine) in fish sauce, which was similar to the report on *Tetragenococcus* in soy sauce [52]. These results also suggest that *Tetragenococcus* may be a good strain to be developed in fish sauce.

Interestingly, at 15 months of fermentation, the bacterial community was the most diverse (Figure 6), which was consistent with the previous diversity index results (Table 2). The relative abundances of *Pseudomonas*, *Pseudoalteromonas*, *Tissierella*, *Carnobacterium*, *Psychrobacter,* and *Gallicola* significantly increased (Figure 6). As a typical cold-tolerant and aerobic bacterium, *Psychrobacter* produces lipase, leading to the hydrolysis of oil and producing a foul odor and musty smell [53]. *Pseudomonas* is a common putrefaction bacterium in the storage process of aquatic products. It produces proteolytic enzymes to rapidly putrefy seafood, thus affecting shelf life [54]. However, in this study, there was no significant positive correlation between these two genera and TMA (Figure 7: TMA), and it was speculated that the increase in oxygen content in the fermentation broth during stirring might lead to the growth of these two genera.

*Pseudoalteromonas* is a gram-negative pseudomonas that is often found in the sea [55]. It has been reported that *Pseudoalteromonas* inhibits *Vibrio* growth to some extent [56], but in this study, *Vibrio* sp. abundance was stable, and there was no significant negative correlation with *Pseudoalteromonas*. It has been reported that members of *Vibrio* might be associated with spoilage or abnormal seafood fermentation. However, in this study, the species abundance of *Vibrio* was relatively stable, and the content was not high (Figure 6). Correlation analysis showed that there was no significant positive correlation between *Vibrio* and TMA (Figure 7: TMA). Therefore, the presence of *Vibrio* had little impact on the quality of fish sauce.

It is worth noting that in the later stage of fermentation, species with abundances less than 1% were classified as “others,” which accounted for an increasing proportion (Figure 6). However, we still know very little about the species information and mechanism of action of these rare microorganisms and their influence on fish sauce fermentation. In addition, the influence of the bacterial community on the flavor of fish sauce and the mechanism of flavor-forming substances need to be further studied.

In conclusion, this study used a high-throughput sequencing method to reveal, to some extent, the bacterial community succession and quality changes in the fermentation process of traditional Chinese fish sauce and the correlation between them. Our results provide new information for the design of tailored starter cultures for TCFS, broadening our knowledge of microbial structure and function in traditional Chinese fermented foods.

## Figures and Tables

**Figure 1 microorganisms-07-00371-f001:**
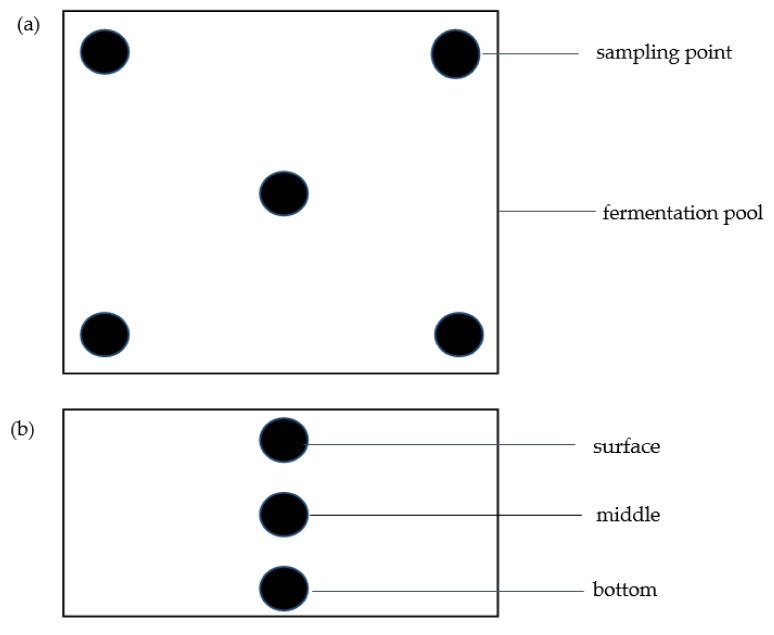
Sampling demonstration diagram of fish sauce fermentation broth: top view (**a**) and front view (**b**).

**Figure 2 microorganisms-07-00371-f002:**
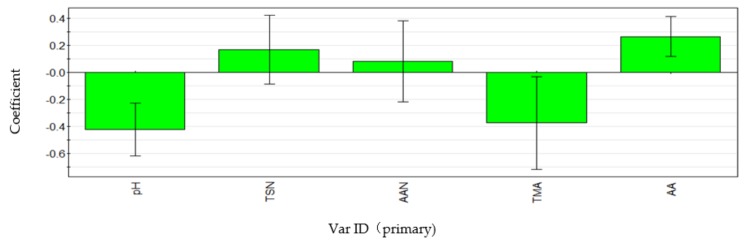
Regression coefficient of fermentation time and physical and chemical index. TMA is formed by reducing trimethylamine N-oxide (TMAO) in the decomposition process of fish and meat due to bacterial action.

**Figure 3 microorganisms-07-00371-f003:**
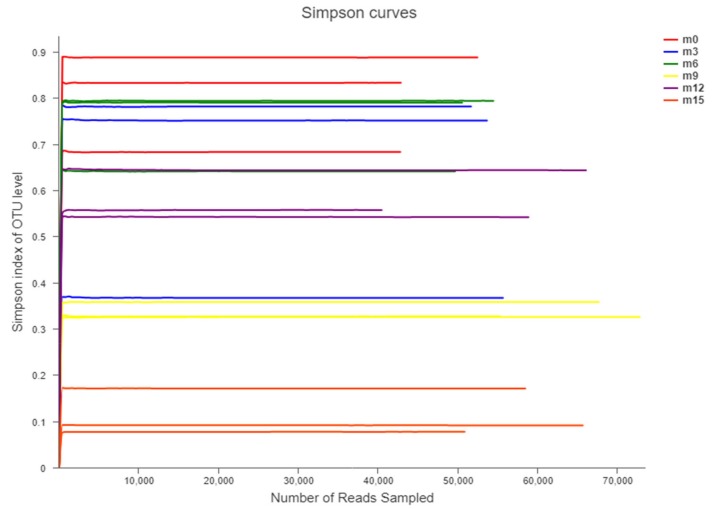
Rarefaction analysis of high-throughput sequencing reads of the bacterial 16S rRNA gene from 18 fish sauce samples. Rarefaction curves were constructed at a 97% sequence similarity cut-off value.

**Figure 4 microorganisms-07-00371-f004:**
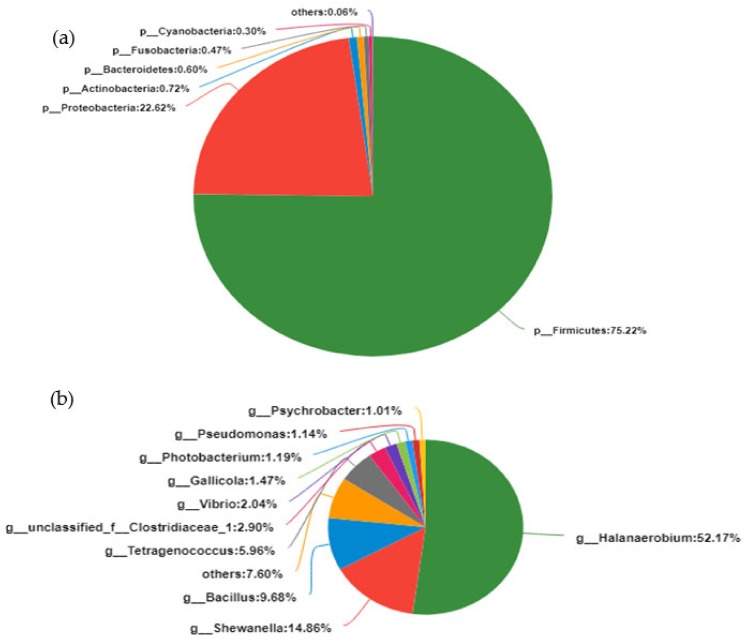
Pie diagram analysis of bacterial structure in the whole fermentation process of fish sauce at the phylum (**A**) and genus (**B**) level.

**Figure 5 microorganisms-07-00371-f005:**
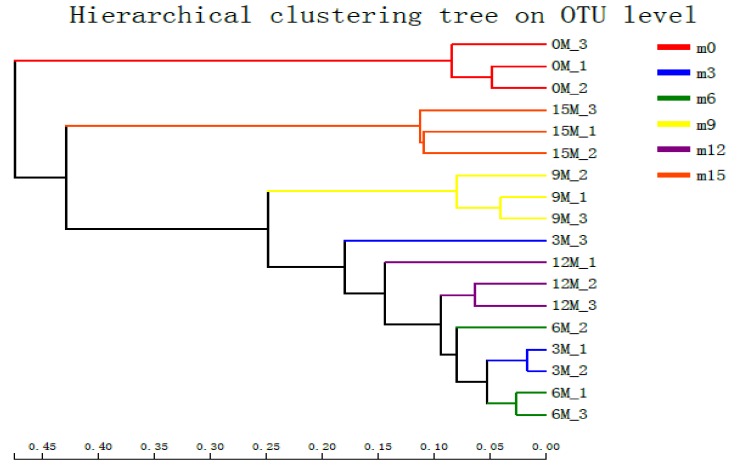
Beta diversity heat map analysis of bacterial communities in fish sauce at different fermentation stages.

**Figure 6 microorganisms-07-00371-f006:**
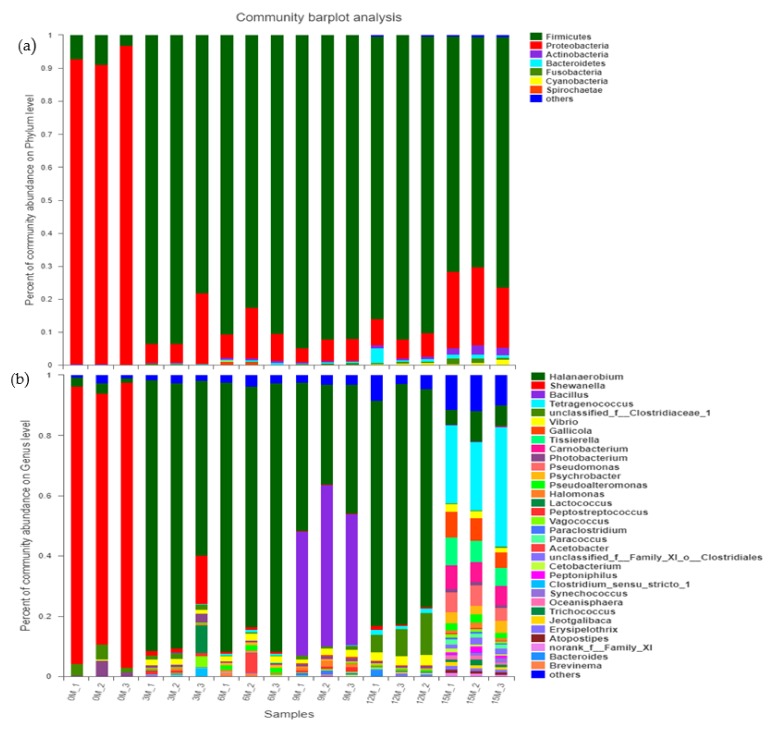
Stacked-column analysis of relative abundances of bacterial communities in fish sauce at different fermentation stages at the phylum (**a**) and genus (**b**) level.

**Figure 7 microorganisms-07-00371-f007:**
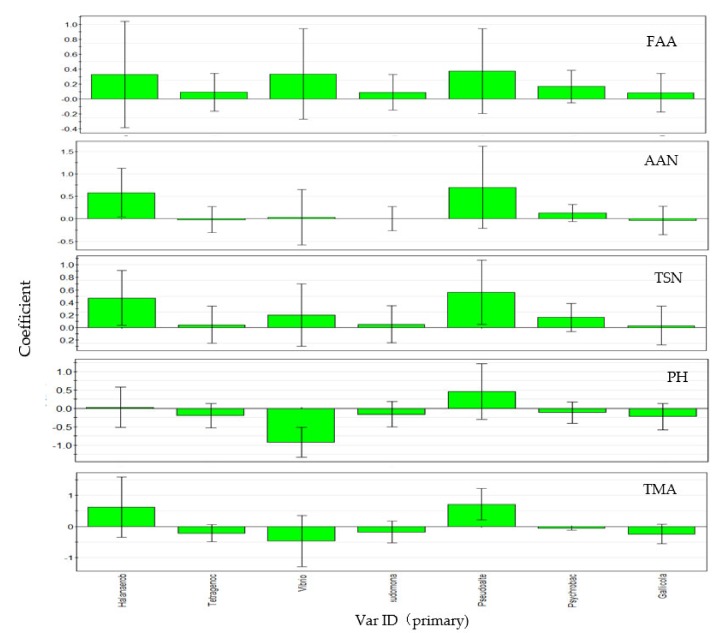
Analysis of the contribution of each index to the quality of fish sauce.

**Table 1 microorganisms-07-00371-t001:** Analysis of the physical and chemical indicators in fish sauce at different fermentation stages.

Time (months)	Average content ± SD				
	pH	TSN (mg/mL)	AAN (mg/mL)	TMA (μg/mL)	FAA (mg/mL)
0	6.2 ± 0.3 ^a^	4.7 ± 0.1 ^d^	2.0 ± 0.1 ^e^	105 ± 5.5 ^a^	4.2 ± 0.1 ^f^
3	5.9 ± 0.0 ^b^	8.8 ± 0.0 ^c^	4.9 ± 0.2 ^d^	140 ± 12.4 ^a^	10.0 ± 0.1 ^e^
6	6.0 ± 0.2 ^ab^	12.1 ± 0.2 ^a^	10.2 ± 0.2 ^a^	140 ± 0.2 ^a^	27.1 ± 0.3 ^b^
9	5.8 ± 0.1 ^bc^	8.8 ± 0.1 ^c^	6.3 ± 0.0 ^c^	111 ± 0.0 ^a^	16.1 ± 0.1 ^d^
12	5.5 ± 0.0 ^d^	9.5 ± 0.2 ^b^	6.1 ± 0.4 ^c^	104 ± 11.6 ^a^	24.2 ± 0.2 ^c^
15	5.6 ± 0.1 ^cd^	12.0 ± 0.1 ^a^	7.7 ± 0.0 ^b^	94 ± 5.8 ^a^	30.6 ± 0.2 ^a^

^a,b,c,d,e,f^ Statistical analysis using a one-way ANOVA (Duncan, Tukey, *p* < 0.05), with same letters showing no significant differences. TMA is formed by reducing trimethylamine N-oxide (TMAO) in the decomposition process of fish and meat due to bacterial action.

**Table 2 microorganisms-07-00371-t002:** 16S rRNA sequencing and diversity data statistics of bacteria in fish sauce during fermentation.

Sample	Seq. Num.	Total OTUs	Chao Index	Shannon Index	Simpson Index
0M_1	44729	133	237	0.480	0.833
0M_2	46051	163	331	0.879	0.683
0M_3	54946	115	194	0.362	0.888
3M_1	53488	217	323	0.781	0.782
3M_2	54578	251	380	0.886	0.751
3M_3	61583	220	415	1.712	0.367
6M_1	55788	350	461	0.778	0.794
6M_2	51342	379	556	1.170	0.641
6M_3	51945	358	513	0.797	0.790
9M_1	70991	346	517	1.577	0.358
9M_2	58086	392	645	1.781	0.327
9M_3	77459	421	692	1.770	0.325
12M_1	41841	458	604	1.625	0.557
12M_2	60149	486	609	1.408	0.542
12M_3	67121	425	559	1.145	0.643
15M_1	68797	708	801	3.559	0.091
15M_2	52977	662	757	3.662	0.077
15M_3	60803	647	730	3.136	0.171

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
