# Peer review of "Dynamic Changes in the Bacterial Community During the Fermentation of Traditional Chinese Fish Sauce (TCFS) and Their Correlation with TCFS Quality"

_microorganisms, 2019, doi:10.3390/microorganisms7090371_

Round 1
Reviewer 1 Report
A revised copy including suggested changes is here attached.

Author Response
The authors are grateful to the Reviewer 1 for their time and patience, and give a positive evaluation to our manuscript. All revisions can be found in revised version.
Are those local names for fish sauce? Please explainResponse: Yes. Chaoshan, Guangdong is one of the sources of fish sauce in China
Please specify what is itResponse: Jia zi is where the fish were caught, which site in Shanwei city, Guangdong province, China.
Please specify what is itResponse: Lu Feng is where the fish were caught, which site in Shanwei city, Guangdong province, China.
It is not clear the volume of tank and the amount of fish+brine inside. How much is a portion? How many are the portions? What is the ratio fish/solar solution? Please clarifyResponse: The capacity of each fermentation tank is 30t, and salt accounts for 30%, so the total amount of fish and salt in each fermentation tank is 21t and 8t respectively.
Do the Authors mean tons?Response: Yes. ”t” mean tons.
How much sample for each point ?Response: The sampling quantity of each point is 0.25kg, and the sampling quantity of each sample is about 4kg.
It could be important to characterise the food ecosystem also the determination of water activity (Aw) and pHResponse: pH was measured, but the significant difference was not obvious compared with other physical and chemical indicators, so it was not included in the article. Now the pH data has been supplemented.
Meanwhile, pH is an important indicator reflecting the process of lactic acid fermentation, and it can be seen in figure 2 that pH presents a declining trend, and it reaches 5.55 at 15m.
Delete, it is a basic information useless for the discussion. Include any consideration about Lactobacillus in this ecosystemResponse: The author added the basic information in order to make the article easier to read and more coherent.
Reviewer 2 Report
In this study, a high-throughput sequencing technology was used to reveal the dynamic changes in the bacterial community during the natural fermentation of TCFS. The article is well written and could have an impact to the research on FFF.
Author Response
In this study, a high-throughput sequencing technology was used to reveal the dynamic changes in the bacterial community during the natural fermentation of TCFS. The article is well written and could have an impact to the research on FFF.
Response: The authors are grateful to the Reviewer 2 for their time and patience, and give a positive evaluation to our manuscript.
Reviewer 3 Report
The paper is well written. The comments are provided below to add more clarity and details to the manuscript
The authors need to provide further details on the significance of this study in the introduction as to how the studying the microbial community structure would help in improving the quality/methods of fish soy sauce (Lines 43-46) Figure 2 graph is not clear Define TMAO as Trimethylamine N-oxide Check Tukey analysis on TMA in Table 1. Both amino nitrogen and TMA first increases and then decreases, why authors did not discuss other chemical indicator changes. Please provide data on pH shift, data on pH change is missing, which is important in these type of fermentation It’s understood the samples from corner and center points and surface, middle and bottom were mixed before analysis. It would be interesting to provide any information as a supplementary data on the variance they have observed at least on the physical and chemical indicator changes. Author needs to explain further on why the changes in microbial community structure were observed only during in between period (3-12 months) as compared to end and start of the fermentation period. Why certain genus of the microbial community was selected for correlation with environmental factors?
Author Response
1.The paper is well written. The comments are provided below to add more clarity and details to the manuscript. The authors need to provide further details on the significance of this study in the introduction as to how the studying the microbial community structure would help in improving the quality/methods of fish soy sauce (Lines 43-46)
Response: The quality and flavor of TCFS are closely related to the microbial community. The core strains of fish sauce fermentation could be selected by defining the rule of bacterial community succession in fish sauce fermentation and analyzing the correlation between bacterial community and fish sauce quality. In particular fermentation stage, beneficial bacteria can be added to accelerate the fermentation process, shorten the fermentation period and improve product quality. It is of great significance to monitor and standardize the traditional technology and the quality and safety of TCFS to clarify the microbial community structure and its changes and functions in the process of fish sauce fermentation.
Figure 2 graph is not clear Define TMAO as Trimethylamine N-oxide Check Tukey analysis on TMA in Table 1.Response: The sentence “TMA is formed by reducing trimethylamine N-oxide (TMAO) in the decomposition process of fish and meat due to bacterial action” was added into the legends of Fig. 2 and Table 1.
Both amino nitrogen and TMA first increases and then decreases, why authors did not discuss other chemical indicator changes. Please provide data on pH shift, data on pH change is missing, which is important in these type of fermentation.Response: pH was measured, but the significant difference was not obvious compared with other physical and chemical indicators, so it was not included in the article. Now the pH data has been supplemented.
Meanwhile, pH is an important indicator reflecting the process of lactic acid fermentation, and it can be seen in figure 2 that pH presents a declining trend, and it reaches 5.55 at 15m.
It’s understood the samples from corner and center points and surface, middle and bottom were mixed before analysis. It would be interesting to provide any information as a supplementary data on the variance they have observed at least on the physical and chemical indicator changes.Response: Considering that the sampling points before mixing cannot represent the quality of the whole sample, the physical and chemical indexes are not determined separately.
Author needs to explain further on why the changes in microbial community structure were observed only during in between period (3-12 months) as compared to end and start of the fermentation period. Response: At the initial stage of fermentation (0-3m), microorganisms were mainly carried by fish and seawater, and salt was not completely immersed in the fish, so the structure of bacterial community did not change much. In the middle stage of fermentation (3-12m), the fish body slowly dismembered and the salt gradually integrated into the fish body, the structure of the bacterial community began to change, and the dominant genus gradually appeared. In the later stage of fermentation (15m), the salt was completely integrated into the fish body, the environment was stable, so the bacterial community structure tended to be stable. Why certain genus of the microbial community was selected for correlation with environmental factors?Response: Because the certain genera of the microbial community selected for correlation with environmental factors were belonged to the dominant genera during the process of fish sauce fermentation, and their changes displayed certain regularity and coincide with the changes of physical and chemical indexes.